# Scale and Drivers of Female Agricultural Labor: Evidence from Pakistan

**Iqra Mohiuddin [1], Muhammad Asif Kamran [2], Shokhrukh-Mirzo Jalilov [3], Mobin-ud-Din Ahmad [3], Sultan Ali Adil [1], Raza Ullah [1] and Tasneem Khaliq [4,\*]**

1   Institute of Agricultural & Resource Economics, University of Agriculture Faisalabad, Faisalabad 38000, Pakistan; iqramohiuddinch@gmail.com (I.M.); sultan.adil@uaf.edu.pk (S.A.A.); raza_khalil@yahoo.com (R.U.)
2   Nuclear Institute for Agriculture and Biology, Faisalabad 38000, Pakistan; agriecon.niab@gmail.com
3   CSIRO Land and Water, Black Mountain Science and Innovation Park, Canberra ACT 2601, Australia; shokhrukh.jalilov@csiro.au (S.-M.J.); mobin.ahmad@csiro.au (M.-u.-D.A.)
4   Department of Agronomy, University of Agriculture Faisalabad, Faisalabad 38000, Pakistan
\*   Correspondence: drtasneem@uaf.edu.pk; Tel.: +92-333-839-7076

**Abstract:** Agricultural labor is largely informal, particularly for female agricultural labor in developing countries. Despite significant participation in the agricultural labor force in Pakistan, women's contribution is not properly acknowledged and rewarded. The issue is further aggravated by the dearth of literature on gender–labor relations in cropping and livestock activities. Considering this gap in the literature, the current study was conducted with the specific objective of exploring the labor composition of different agricultural activities in different farm size categories in general and, particularly, female agricultural labor (family and hired labor) participation and its determinants in the rice–wheat cropping system of the Punjab province, Pakistan. The data were collected from 300 households across four districts of the province. Labor participation was calculated on an official farm size classification basis, i.e., small (<12.5 acres), medium (12.6–25 acres) and large (>25 acres) farms. The findings show that female labor is predominantly demanded in the manual harvesting of wheat, rice nursery transplantation and harvesting, and the majority of the livestock-related activities. The regression model results showed that family female labor and hired female labor participation significantly depend on the landholding status of farmers, household size, family type and level of education. The interviews also illustrated that labor relations are rapidly changing—ongoing mechanization threatens conventional female labor activities due to the lack of machinery operation skills among females, caused by informal state policies and cultural barriers. The findings of the study have important policy implications for mainstreaming gender status in agricultural policy and rural development and contribute directly to the Sustainable Development Goals on Gender Equality (SDG#5) and Decent Work and Economic Growth (SDG#8), and indirectly to No Poverty (SDG#1), Zero Hunger (SDG#2), Responsible Consumption and Production (SDG#12) and Climate Action (SDG#13).

**Keywords:** female labor; rice-wheat system; farm size; agricultural labor; Pakistan; gender bias

## 1. Introduction

Historically, women remained disempowered socially and economically due to the discriminatory policies of internal institutions and organizations [1]. The proportion of unpaid rural male and female labor has been reported as 17% and 60%, respectively [2]. The Organization of Economic Cooperation and Development (OECD) defines unpaid labor as time spent doing routine housework, shopping for

necessary household goods, childcare, tending to the elderly and other household or non-household members, and other unpaid activities related to household maintenance. While women are participating in all agricultural activities from sowing to the post-harvesting of crops and management of livestock [3], the gender discrimination, labor terms and conditions and low wages for female work reduce women's incentives to fully participate in agricultural practices, and in turn create insufficiencies in human capital investment and productivity [4,5]. Female participation in agricultural activities in Southeast Asia shows country-specific variations in gender equity and empowerment. [6].

The agricultural labor market is relatively unorganized and informal compared to the industrial and service sectors in many countries. Pakistan, being an agricultural country, has agriculture as an integral part of the national economy, with an 18.5% share in the gross domestic product (GDP). Livestock makes up 60.5% of the share of agriculture, translating to 11.2% of the GDP [7]. Agriculture and livestock products provide food security and livelihoods to 208 million people [2].

More than 50% of the population and 42% of the labor force of Pakistan resides in rural areas [8]. Pakistan's agriculture is labor intensive; women make an essential contribution to it and their roles are substantially different by region and are changing rapidly in different areas. Despite their active participation in the farm sector, women have less access to assets, services and opportunities compared to men. This gender gap is found generally for access to inputs, services, land ownership, livestock, technology, education, extension and financial services. Empirical evidence shows that if women were to have the same access to productive resources (machinery, market, stakeholders, etc.) as men, they could increase the yields on their farms by 20–30% [8].

The female labor contribution in livestock is well documented and is recognized [9]. In Pakistan, too, women play an important role in different livestock activities like fodder cutting, fodder chopping, watering and feeding of animals, shed cleaning, milking, preparing dung cakes and looking after the health of the herd [10]. Rice and wheat are the dominant food crops of the country, which attract both male and female labor during sowing and at harvesting, i.e., peak crop activity periods [11].

Due to increased use of mechanization and the migration of the male members from rural to urban areas, the role of rural women is changing from unpaid family labor to farm managers, a phenomenon termed the "feminization of agriculture". Rural women are now actively involved in agricultural activities such as rice and wheat cropping. When a rural woman works outside, she can contribute to the family income, which has a great impact in improving their livelihood and status. In fact, this change can help them to also take part in family decisions, which indicates the empowerment of rural women. Despite the fact that the greater role played by women in wheat cultivation in Nepal is shifting their status from farm laborers to managers, their contribution is still undervalued as men are targeted for state-driven advisory and training-related support services [12]. However, the contribution of women has not been documented as much as it should be. The research results from different dimensions depict that the overall position of the selected women ranged between low to medium and that they do not have a separate wage market in rice wheat cropping.

The existing literature on agricultural labor is patchy, with limited information about labor input in the production process in general and about activities performed by female laborers. For example, Ref. [13] analyzed a sample of 363 households collected from four villages in Iran and found positive attitudes from female farmers towards agriculture sustainability. Another study collected data about agricultural labor from 500 farmers in South Punjab, Pakistan and found that the majority of livestock activities were performed by female laborers [14]. Social and cultural barriers hinder effective female participation in agriculture, and other economic activities were investigated via the resistance to education and other learning activities [15]. The economic literature on female choice to stay out of jobs in favor of domestic chores explains this through the logic of the lower opportunity cost for market activities, intra household welfare consideration and work dynamics [16]. Female access to productive and decent work is possible by removing religious and cultural constraints, efficient public provision of necessary services to reduce the time spent on household activities and access to skills enhancement opportunities to enter the job market [17]. While there are several other studies on gender–labor

participation and socioeconomic barriers, there is a dearth of literature on gender–labor relations in families and hired labor and its determinants. This current study, in this context, provides empirical evidence of the scale and drivers of female labor participation by:

- identifying activities dominated by family and hired female labor in the rice-wheat cropping system of the Punjab province;
- analyzing gendered family and hired labor force dynamics across different farm scales;
- determining drivers of family female and hired female labor in farm activities.

## 2. Conceptual Framework

Due to the lack of secondary data and any detailed study about agricultural labor segregation on a gender basis, a Participatory Rapid Appraisal (PRA) was done before conducting a detailed data collection. The objective of the PRA was to understand the context and labor participation in agricultural activities and to develop a detailed questionnaire to collect the data. Based on the PRA, a conceptual framework was developed to understand the agricultural labor dynamics. The conceptual framework presented in Figure 1 shows the linkages between agriculture labor and how farm size influences the use of family and hired labor.

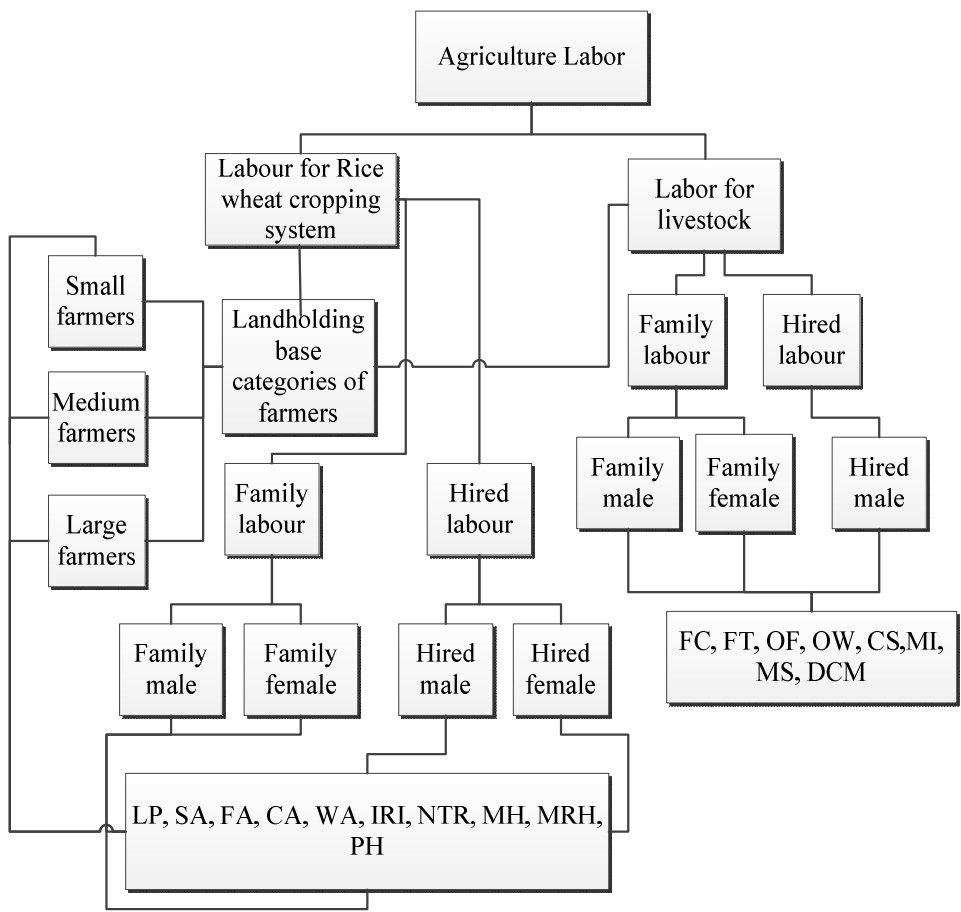

**Figure 1.** Conceptual framework of labor participation in rice–wheat cropping system and livestock activities in Punjab province (abbreviations are explained in Table 1).

Agricultural labor has two major categories: crop and livestock labor. Both categories have a labor composition made up of hired and family labor (male and female) which are categorized on the basis of farm size. Table 1 provides the list of common cropping and livestock activities and their related abbreviations in this study.

**Table 1.** Abbreviations used in the conceptual framework.

| Abbreviation | Full Form | Abbreviation | Full Form |
| --- | --- | --- | --- |
| LP | Land preparation | PH | Post-harvesting |
| SA | Seed application | FC | Fodder cutting |
| FA | Fertilizer application | FT | Fodder transportation |
| CA | Chemical application | OF | Offering fodder |
| WA | Weedicide application | OW | Offering water |
| IRI | Irrigation | CS | Cleaning of shed |
| NTR | Nursery transplantation of rice | MI | Milking |
| MH | Manual harvesting | MS | Milk selling |
| MEH | Mechanical harvesting | DCM | Dung cake making |

Based on the PRA, labor was categorized into family labor and hired labor among both the male and female genders. It shows that both family and hired labor participation depend on the landholding categories of farmers (small, medium and large) in cropping activities. The family and hired labor participation in all cropping activities and livestock activities for males and females have been estimated. Official land holding classifications were used for the comparison of three farm sizes, namely small (up to 12.5 acres ~5 hectares), medium (12.6 to 25 acres ~5–10 hectares) and large farms (above 25 acres ~more than 10 hectares). The conceptual framework provides an overview of the labor composition, while the numerical values and analysis are given in the Results and Discussion section.

## 3. Methods and Data Analysis

### 3.1. Study Area and Data Collection

Rice-wheat cropping is the dominant cropping system of the Sindh and Punjab provinces of Pakistan. As the Punjab has a major cropping system and also has rainfed and irrigated areas in this system, data were collected from four major rice-wheat zone districts of the Punjab: Gujranwala, Hafizabad, Sheikhupura and Narowal. Seventy-five farmers were randomly selected from each district, giving a total sample of 300 respondents. The location of the selected study area is presented in Figure 2.

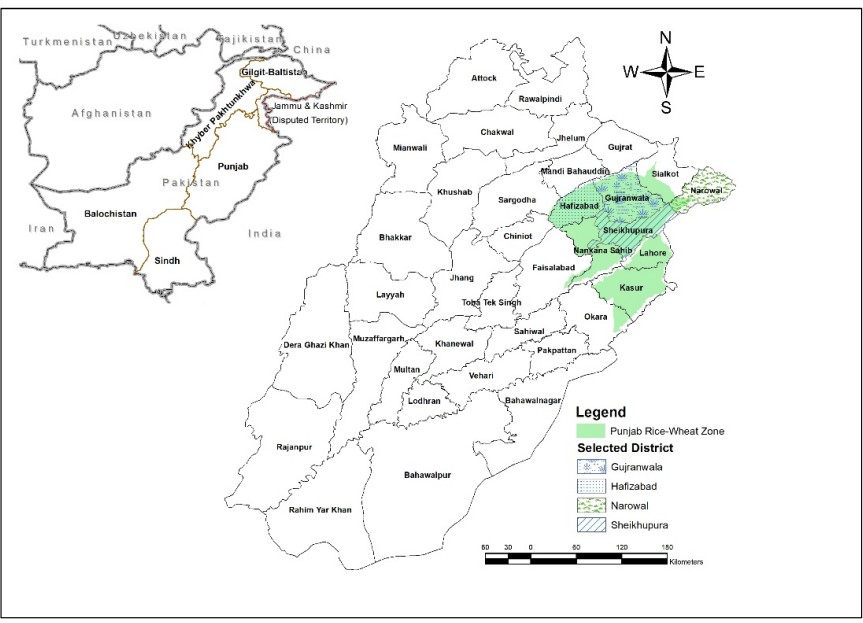

**Figure 2.** Study area.

Before engaging in a full-scale survey in the selected sites, researchers did an extensive desk study of previous studies' techniques and analyses in addressing the inevitable challenges. The biggest challenge was how to appropriately invite female respondents to participate. In Pakistan, like other Islamic countries, no stranger can approach unknown women for interview without consent from their father, brother or husband. As one measure, the team hired experienced female enumerators who already had experience of such field work, as the team encountered cases where women refused to be a respondent if a man was the enumerator.

After drafting the questionnaire, the team conducted a day-long trip to conduct the preliminary survey in Saboki village, Gujranwala district in December 2018. The main objective of this visit was to test the pilot questionnaire and make necessary changes and improvements if needed. The team received useful feedback in addition to invaluable discussions with female and male groups separately. The initial opinion of the researchers about female labor in agriculture was that the wage data would be easily available for different tasks. However, due to the informal nature of the agricultural labor market and multiplicity of tasks performed, the questionnaire was later modified to indirectly measure wage and participation based on proportionate participation in different tasks. Based on that feedback, the questionnaire was revised and updated. The full-scale survey was conducted in April–July 2019. Each enumerator was assigned a predefined geographic location where s/he was in charge of face-to-face interviews. Again, female enumerators handled interviews with females and male enumerators handled male respondents. The data were collected from 75 female and 225 male respondents. The number of female respondents is less because the majority of the randomly selected households either did not have family females participating in farming or the relevant female was not aware of managerial decisions and female labor hiring-related issues. However, to ensure objective responses, the female enumerators triangulated the female labor-related responses and doubly verified them with both the male and female household members. The biasedness of the response in gender-sensitive studies was known to the researchers and, consciously, both male and female family members were engaged in labor-related questions to handle the issues of biased responses and female representativeness appropriately.

*3.2. The Model*

The nature of farm labor, i.e., family and hired as well as the gender of the family and hired laborers, was also taken into account and compared using Analyses of Variance (ANOVA). Tukey's HSD ("honestly significant difference") test [18] was used for post-hoc analyses, because the comparison of the categories of famers for labor participation was also done using Tukey's HSD test. Multiple linear regression (MLR) was used to model the linear relationship between the explanatory (independent) variables and response (dependent) variable. In the first model, total female labor is estimated using the following equation:

$$\text{FemLab} = \beta_0 + \beta_1 \text{ FSize} + \beta_2 \text{ HSize} + \beta_3 \text{ FType} + \beta_4 \text{ Educ} + \mu \ldots$$

where:

FemLab = Female labor participation (both family labor and hired labor) in rice–wheat cropping system;
FSize = Farm size (acres);
HSize = Household size (in numbers);
FType = Family type of respondents (joint or nuclear);
Educ = Level of education of respondent (schooling years);
$\mu$ = Disturbance term or error term.

Table 2 describes expected signs of the independent variables for hired and family female labor participation. The expected negative relationship for family females and the positive relationship for hired females with landholding size, household size and education can be attributed to the fact that

larger landholdings require the increased participation of hired labor as family labor is not sufficient for the needs of larger landholdings. Similarly, as the household size increases, less family female labor is expected to participate in farming activities due to the availability of male members in a country with dominant small farm sizes. As the level of education of the household head increases, there is a greater possibility that an educated household will encourage household members to pursue education instead of farming and, therefore, it is hypothesized that female family labor participation will go down. However, to compensate for the lack of female laborers to carry out specific tasks, households with a greater education level and a bigger household size may hire more female laborers. In nuclear families, there is a higher likelihood of family female laborers, as the female is solely responsible for household chores, but in joint families there is an expected positive relationship with family female labor.

**Table 2.** Variable descriptions and expected signs with hypothesized relationship.

| Independent Variables | Measurement Units | Dependent Variables | | Hypothesized Relationships |
| --- | --- | --- | --- | --- |
| | | Family Female | Hired Female | |
| Land holding | Land in acres | – | + | Higher landholding generally results in bigger land–family labor ratio and therefore reduces the per acre family labor supply and increases the hired labor ratio to fulfil labor demand |
| Household size | Number of family members | – | – | Families with larger household sizes do not generally prefer to engage females in the family in agriculture; instead of this, they hire female labor. |
| Family type | Joint = 1 Nuclear = 0 | + | – | Joint families have common land to work and strong family ties to till the land. |
| Education of household head | Number of schooling years | – | + | An educated household head prefers to educate members to enable them to seek better wages and non-farm employment instead of farm activities. |

## 4. Results and Discussion

### 4.1. Overview

The respondents had different holdings, i.e., small, medium, and large. Land holding information helps us to understand farm size distribution of respondents and linked factors of labor and mechanization. Table 3 shows that the majority of respondents are farmers with small and medium farms (96%), with a small number of large farms (4%). Regarding labor, the data show that family male labor has a higher participation (49.6%) in rice–wheat cropping compared to family female labour, hired male and hired female labor.

**Table 3.** Socioeconomic and general farming-related information.

| Variable | Numbers | Percentage |
| --- | --- | --- |
| Farm Category | | |
| Small Farmers (up-to 12.5 acres) | 231 | 77 |
| Medium Farmers (12.6–25 acres) | 57 | 19 |
| Large Farmers (above 25 acres) | 12 | 4 |
| Total | 300 | 100 |
| Harvesting of Rice Crop | | |
| Manual | 98 | 33 |

**Table 3.** *Cont.*

| | | |
|---|---|---|
| Combine Harvester | 202 | 67 |
| Total | 300 | 100 |
| Harvesting of Wheat Crop | | |
| Manual | 107 | 36 |
| Tractor Reaper | 2 | 0.7 |
| Combine Harvester | 191 | 63.3 |
| Total | 300 | 100 |
| Male and Female (Mean Labor Hours) | | |
| Family Male | 606 | 50 |
| Family Female | 131 | 11 |
| Hired Male | 427 | 34 |
| Hired Female | 59 | 5 |
| Total | 1223 | 100 |

The data reveals that there is a large proportion (77%) of small farms in the districts under study. The data show that, due to the high proportion of small farmers, there is a 49.5% participation of family labor in rice wheat crop activities such as land preparation, seed application, fertilizer application, chemical application, irrigation, manual harvesting, mechanical harvesting and post-harvesting. Crop harvesting status shows that the majority of the farmers are mechanically harvesting with a combine harvester.

The remainder of the paper is designed to discuss the labor participation in different crops and livestock activities on the basis of family and hired labor in relation to gender, labor composition, farm size classification (i.e., small, medium, large farms), and determinants of female labor (family as well as hired) in different agricultural activities in the study area.

*4.2. Farm Labor Composition by Gender and Hiring Status*

There is a clear distinction in agricultural labor activities based on gender. PRA results showed that men are conventionally engaged in physically demanding activities and continue to remain in those activities even now that these have become much easier due to mechanical interventions like seed bed preparation and threshing. Similarly, the conventional female labor activities have been replaced by machines and, as men traditionally dominate machine operation, they are duly supported by state policy to be provided with government agricultural machinery training, with centers dominated by men and de facto meant to train men for the operation of driving-based and other machinery. This has disadvantaged female labor and reduced women's bargaining position, as female labor is now limited to relatively harder manual tasks compared to men, who handle mostly machine-driven and better paid agricultural labor operations. The results, therefore, are in agreement with this observation and reveal that the majority of agricultural labor in wheat crop activities is conducted by male participants compared to female laborers. The majority of the activities such as fertilizer application, chemical application and irrigation are customarily male-dominant activities, while manual harvesting and post-harvesting are the only activities in the cropping system that are performed by women. As reported by [19], females are not participating in the mechanical harvesting of crops due to a lack of skills and awareness of the use of technology on the farm. As farm mechanization advances, and it is likely to replace labor, it is highly likely that females will be forced into less lucrative and more physically demanding tasks if they are not equipped with the skills to handle mechanical devices and technology in agriculture. Interestingly, women are now carrying out more physically demanding activities compared to men.

Figure 3 describes the participation of male and female laborers in terms of hired and family laborers in wheat cropping activities. It shows that hired and family male participation is higher than female in all activities of wheat cultivation. Females are participating only in manual and post-harvesting activities. However, the participation of hired females is comparatively higher than family females. The table shows that land preparation, seed, fertilizer and chemical application, irrigation and mechanical harvesting are purely male labor activities in wheat cropping. It is generally believed that activities that require hard physical labor or nighttime engagement (irrigation, due to its rotation across farms) are conventionally male activities. As the mechanization trend is increasing, the conventionally physically demanding activities are done with relative ease through mechanization, but female entry into these jobs is still restricted. One plausible reason for it could be path dependence and cultural restrictions on women driving in rural areas. Now, as harvesting and threshing is increasingly performed mechanically, women's labor in these activities is also declining.

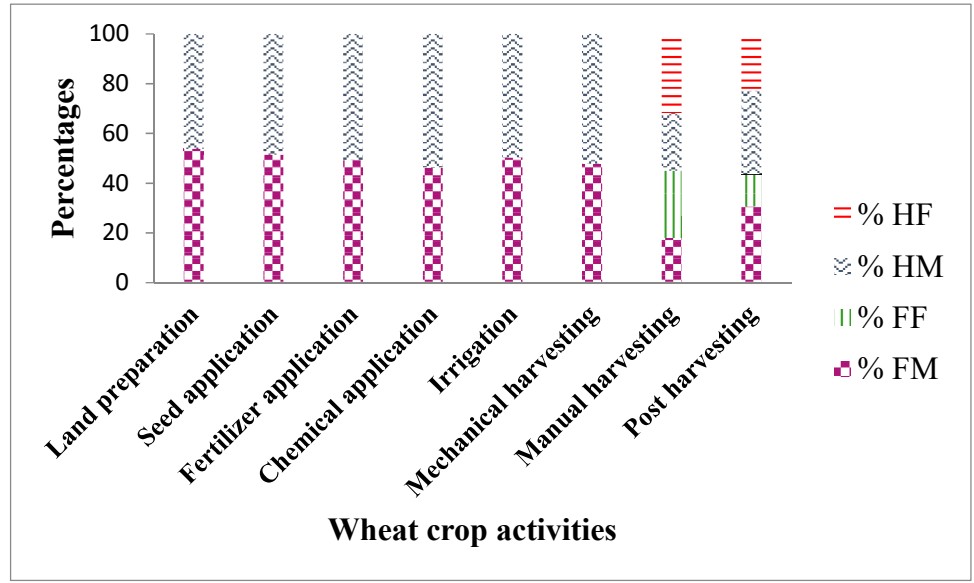

**Figure 3.** Labor participation in wheat cropping activities.

In contrast to wheat cropping, rice cropping is relatively more dependent on manual labor. The participation of labor for rice nurseries and rice crops has been separately identified because, in both stages of rice sowing, the activities are performed in different phases. In the rice nursery, labor participation starts from the activities of land preparation to nursery transplantation and land preparation to post-harvesting activities. As expected, land preparation and seed and chemical application are male-specific activities because of similar reasons discussed in the wheat crop labor discussion. However, Figure 4 explains that the participation of females is dominant in the nursery transplantation of rice. Males do not work with females for transplanting purposes and it is considered a female activity.

Besides nursery transplantation, female labor plays an active role in the harvesting, threshing and storage of rice. Figure 5 shows the second phase of the rice crop after transplantation and has a similar trend as the wheat crop, which male labor dominates in most of the activities.

Figure 5 shows that the participation of hired and family labor have the same trend in rice as in the wheat crop. Because mechanical harvesting and sowing is replacing the labor in rice and wheat cropping, female participation is being reduced day by day. The PRA results and discussions with experts showed that 20% of rice is directly seeded and mechanically harvested. Moreover, about 80% of wheat in the study area is harvested using a combine harvester. This shows the decrease in female agricultural labor in conventional tasks and the loss of job opportunities due to the lack of access and cultural barriers for women, preventing them from learning to run agricultural machinery.

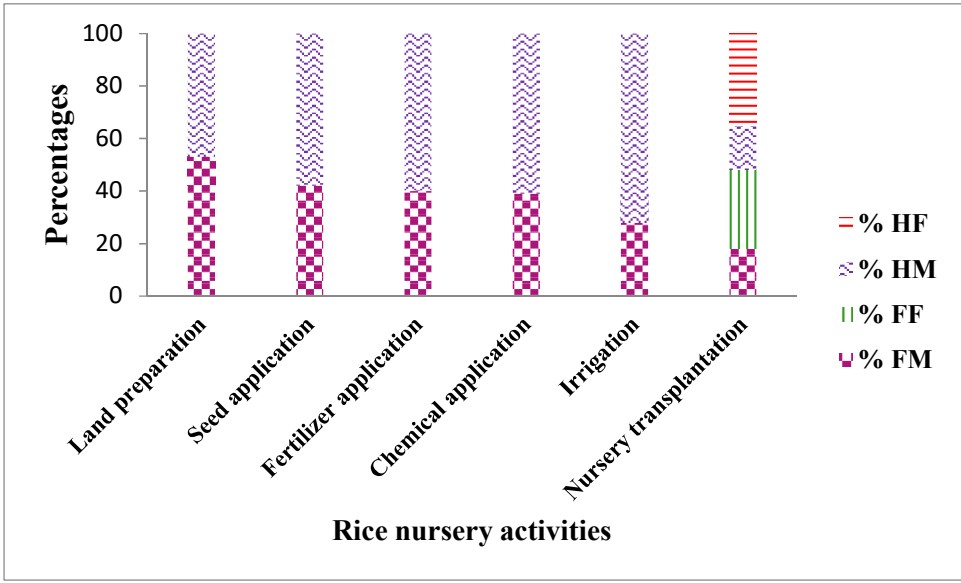

**Figure 4.** Labor participation in rice nursery activities.

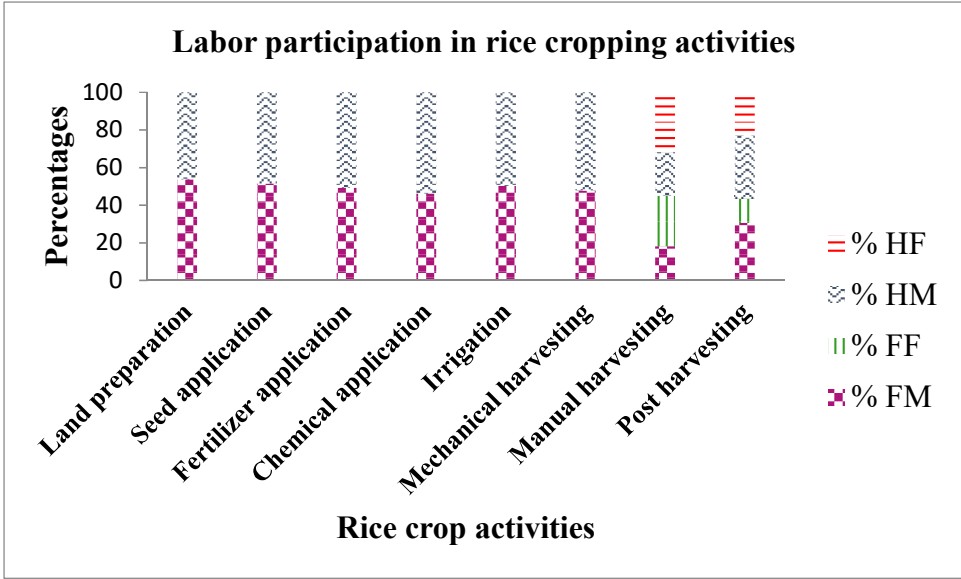

**Figure 5.** Labor participation in rice cropping activities.

Livestock in the area predominantly provides subsistence for family needs and some cash income through the sale of livestock products in the selected cropping zone. It is important to include livestock in the study, as agricultural labor involves performing different tasks and livestock is one of the major labor-intensive activities besides growing crops. Moreover, female labor plays a crucial role in livestock activities as shown in Figure 6.

Figure 6 shows that the majority of the farmers have small landholdings and keep their animals at their own homes. Therefore, men go to the field for cropping activities and female family members perform the job of female labor for the livestock, along with the household chores. Due to this reason, female participation is higher in livestock practices. Family female participation is comparatively higher in offering fodder and water to animals, the milking of animals and the selling of milk. In fodder cutting and transportation, female participation is comparatively lower. The cleaning of the shed is a female-dominant activity. This description represents the fact that the majority of livestock activities are performed by females compared to the males.

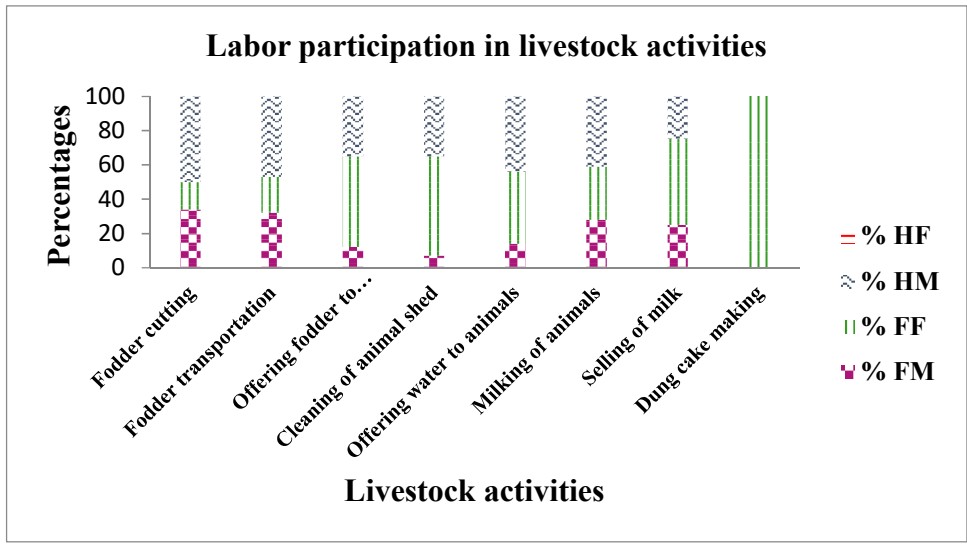

**Figure 6.** Labor participation in livestock activities.

*4.3. Family and Hired Labor Trend across Farm Size Categories*

The village areas have landed and landless classes. The landless class is mostly hired for labor, while family members also perform different labor operations as part of family labor. This section will discuss family and hired labor engagements in different farm size categories.

4.3.1. Family Labor Engagement across Different Farm Sizes

Besides the overall female labor participation in crops and livestock, it is important to understand the mean comparison using Tukey's HSD test. The results of the comparison of labor participation on the basis of landholding category are given in Tables 4 and 5.

**Table 4.** Participation difference of labor among farm categories.

| Mean Comparison | Sum of Squares | Df | Mean Square | F | Significance |
|---|---|---|---|---|---|
| Between Groups | 8.546 | 2 |  |  |  |
|  |  |  | 4.273 |  |  |
| Within Groups | 2818.09 | 1938 |  | 3.985 * | 0.02 |
|  |  |  | 1.454 |  |  |
| Total | 2826.640 | 1940 |  |  |  |

Note: * indicate parameter significance at $\alpha$ = 10%, respectively.

**Table 5.** Mean comparisons for family labor by Tukey's HSD test.

| Farm Categories | Mean Difference | Std. Error | Significance | 95% Confidence Interval Lower Bound | Upper Bound |
|---|---|---|---|---|---|
| Small-Medium | 0.04119 | 0.07619 | 0.05 | −0.1375 | 0.2199 |
| Small-Large | 0.44102 | 0.1827 | 0.04 | −0.0088 | 0.8732 |
| Medium-Large | 0.39983 | 0.19479 | 0.01 | −0.0571 | 0.8567 |

The results reflects that there is a significant difference between groups and within groups for family labor hours in the categories of small, medium and large farms. To know the labor intensity among the farm sizes, a post-hoc analysis (Tukey's HSD test) was used for the ANOVA. The results of the Tukey HSD test are provided in Table 5.

The results in the above table also reveale that there is a significant difference in mean family labor engagement across different farm sizes. The positive value of mean differences shows that by moving

to a larger farm size, the participation of family labor hours relatively decreases. The results are in accordance with our stated theoretical assumption that as the farm size increases, there is a higher likelihood of using machinery on the farm and also hired labor.

### 4.3.2. Hired Labor Engagement across Different Farm Sizes

As hired labor plays a crucial role in crop and livestock activities, this section is dedicated to understanding of hired mean labor engagement per unit area across different farm sizes. In Table 6, the data for hired labor is analyzed across small, medium and large farm size categories. Table 6 depicts the between- and within-group comparisons for the overall significance of the hypothesized relationship.

**Table 6.** Participation difference of labor among farm categories of farmers for hired labor.

| Mean Comparison | Sum of Squares | Df | Mean Square | F | Significance |
|---|---|---|---|---|---|
| Between Groups | 22.041 | 2 | | | |
| | | | 11.021 | | |
| Within Groups | 1048.313 | 916 | | 9.630 *** | 0.01 |
| | | | 1.144 | | |
| Total | 1070.355 | 918 | | | |

Note: *** indicate parameter significance at $\alpha$ = 1%, respectively.

The results reflect that there is a significant difference between groups and within groups for hired labor for the categories of small, medium and large farms. The overall results show that hired labor participation is significantly correlated with a larger landholding category. As the test results do not capture the direction and extent of differences in means, a post-hoc analysis (Tukey's HSD test) is used after the ANOVA. The results of the Tukey HSD test are provided in Table 7.

**Table 7.** Mean comparisons for hired labor by Tukey's HSD test.

| Farm Categories | Mean Difference | Std. Error | Significance | 95% Confidence Interval | |
|---|---|---|---|---|---|
| | | | | Lower Bound | Upper Bound |
| Small–Medium | −0.25475 | 0.7970 | 0.004 | 0.0677 | 0.4418 |
| Small–Large | −0.3909 | 0.15733 | 0.034 | −0.7624 | −0.0238 |
| Medium–Large | −0.64784 | 0.16546 | 0.000 | −1.0363 | −0.2594 |

The analysis reveals the statistically significant differences between small and medium farms as well as between medium and large farms. The minus sign before the mean differences shows the statistical significance within the three categories. The minus sign before the mean difference values represents the fact that small farms have less participation of hired labor compared to the medium and large farms and that medium farms have comparatively less participation of hired labor than large farms. However, there is a comparatively high difference between the small–large and medium–large group. This shows the affordability hired labor, which means that large farms can easily afford hired labor compared to small and medium farms and medium farms can easily afford hired labor compared to small farms; therefore, the mean differences in this table have statistically significant differences.

### 4.4. Factors Influencing Family Labor and Hired Labor Engagement in Agricultural Activities

In the first part of the Results section, we discussed the overall labor composition by considering gender and family or hired labor criteria. After understanding the labor composition for rice and wheat crops and livestock, we then wanted to understand the mean differences in family and hired labor participation across different farm sizes, i.e., small, medium and large farms. We wanted to conclude the labor analysis with a specific analysis of family and hired labor participation using a regression analysis. The independent variables, as mentioned in the hypothesized relationship in the Methods section, include landholding category, household size, family type and education. Table 8 depicts the results from the regression analysis.

**Table 8.** Factors determining female labor participation.

| Variables | Coefficient | Std. Error1 | t-Ratio2 | *p*-Value3 |
|---|---|---|---|---|
| Constant | 1.111 | 0.452 | 2.458 | 0.000 |
| Landholding | −0.140 | 0.239 | −0.874 | 0.03 |
| Household size | −0.217 | 0.101 | −1.924 | 0.05 |
| Family type | 0.326 | 0.180 | 2.717 | 0.008 |
| Education | −0.320 | 0.184 | −1.869 | 0.05 |

Family female labor (hours) participation = 1.111 − 0.140(LH) − 0.217(HS) + 0.326(FT) − 0.320(Edu).

In the equation, the minus sign before the coefficient of landholding status tells us that the family female labor participation decreases −0.140 h when the landholding increases by 1 acre. Therefore, family female participation decreases with increased landholding.

The minus sign before the coefficient of household size represents that, by increasing the size of the household, the family female participation decreases by −0.217%. This is because, by increasing the size of the household, the numbers of males in a family also increase and land to labor ratio squeezing can be a plausible reason for the decline in family female labor participation. The family type has a positive effect on the participation of females, as joint families have more opportunities for females to participate in cropping activities because they have more members in the house to take care of their children. This represents that family type is the most effective factor for family female participation. The negative value of the coefficient of the education level of the household's head indicates that family female participation decreases by 0.32% when the level of education of the head of the household increases (in years). The possible reason for this relationship is that more schooling years tend to keep family females away from family farm labor.

While understanding the factors responsible for family female labor, it is equally important to understand the factors influencing hired female labor in farming activities. The model includes the same variables to understand the engagement of hired female labor (hours) in farming activities (Table 9).

**Table 9.** Linear regression model results of coefficients for hired females.

| Variables | Coefficient | Std. Error1 | t-Ratio2 | *p*-Value3 |
|---|---|---|---|---|
| Constant | 2.791 | 0.282 | 9.904 | – |
| Landholding category | 0.026 | 0.119 | 0.031 | 0.05 |
| Household size | 0.03 | 0.061 | 0.033 | 0.04 |
| Family type | −0.06 | 0.123 | −0.086 | 0.1 |
| Education level | 0.063 | 0.088 | 0.072 | 0.03 |

Hired female labor (hours) participation = 2.791 + 0.026(LH) + 0.03(HS) − 0.06(FT) + 0.063(Edu).

In the equation, the positive sign of the coefficient of landholding status tells us that the hired female participation increases by 0.026% when the land holding increases by I acre. Therefore, hired female participation in crop activities increases. The positive sign before the coefficient of household size represents that, by increasing the size of the household, the hired female participation increases by 0.03%. This is because, by increasing the size of household, the numbers of males in a family also increase, and therefore the family female participation is reduced, but the need to hire female laborers for the activities in which female participation is necessary increases.

The family type has a negative effect on the participation of females. The likelihood of female participation decreases by 6% due to the effect of family type. The positive value of the coefficient of education (schooling years) indicates that female participation increases by 0.063% when the level of education increases.

The finding that female labor contribution is considerably less in field activities is consistent with the [20] observation that, in South Asia, fewer females work in the field. The finding about the

prevalence of female labor in livestock activities is consistent with the findings of [10]. The study's findings about employing family females or hired female laborers for larger landholding sizes can be linked to the findings of [21], in that female laborers are employed in agriculture when male laborers are exhausted and larger farm sizes are the first to exhaust male labor due to the wide family male labor to land ratio. The role of women in the agricultural activities, from seeding to the post-harvesting of crops and the management of livestock, has been reported by [19]. However, that study did not provide exact measures of time dedicated to each activity, as provided by the current study. The role of education in female labor participation in the agriculture sector has not been previously investigated in detail. The existing literature on education and female labor force participation in agriculture is non-existent. The study results reflect the fact that the relationship is positively correlated as far as hired female labor participation and education are concerned. The results are consistent with studies exploring the relationship between education and labor force participation [22–24]. A possible explanation for the positive role of education is that it boosts female confidence to work on farms as hired laborers. However, the negative correlation between family female labor and education could be due to intra-household labor dynamics, resulting in the non-participation of the more educated females in the family compared to less educated females in the family.

## 5. Conclusions

Crops and livestock are major sources of rural employment in the study area. Considering the informal nature of agricultural labor, there are no systematic data available on the composition of agricultural labor on the basis of gender or its contractual arrangement. Primary data collection from 300 households was possible through random sampling from selected districts in the rice–wheat zone in the Punjab, Pakistan. PRA was conducted to understand the agricultural labor dynamics and the nature and extent of mechanization influencing conventional labor relations. The majority (77%) of respondents had small farms (up to 12.5 acres), 19% of farmers had medium farms (12.6 to 25 acres) and only 4% of the respondents had large farms (more than 25 acres). The majority of the respondents who carried out rice (67%) and wheat (63.3%) cropping mechanically harvested their crops with a combine harvester, while 33% and 36% of respondents manually harvested rice and wheat crops, respectively.

The Punjab government should also take steps towards female empowerment in agriculture, as the Sindh government have approved "The Sindh Women Agricultural Women Bill 2019" to provide protection and ensure rights for females engaged in agricultural activities. As mechanization is mostly replacing female laborers in conventionally female-dominant activities and the female laborers are being pushed towards low-paid and more demanding tasks, it is recommended that females should be trained in machine operation; the government should link machinery loans for females with driving licenses to promote better paid and decent jobs for women in agriculture. A future study may look into the rate of labor substitution by mechanization and the differences in expenditure for different operations with manual labor and machinery to enhance our understanding about the future agricultural job market and the desired skills for rural females to remain relevant in the farming job market.

Family male laborers have a major role in crop and livestock activities, followed by hired male laborers. Various factors such as the landholding size of farms and household sizes have negative effects on female participation. The results of Tukey's HSD test show statistically significant differences among three farm size groups for family labor participation. The results show that family labor participation is high among small farms, followed by medium and large farms.

Our regression analysis shows that female labor involvement (both family and hired female) has a statistically significant relation with landholding, household size, education of household head and household type. The results show that men monopolize the mechanized work such as ploughing with a tractor, threshing, harvesting and any other mechanical task; as a consequence, female participation is reduced. Therefore, the government should provide training to rural females through existing infrastructure. Moreover, farm machinery subsidies could be given to females holding licenses to

operate machinery to encourage the female workforce into better paid skilled jobs. This will help female agricultural labor to bridge the wage gap and achieve gender equality (SDG#5).

The majority of the hired labor is landless and small farmers face poverty. To reduce and overcome this poverty, it is necessary that female work should be equally recognized and paid compared to male labor. As advised by the Agriculture Department of Punjab, about 93% of women in the Punjab do not own land, half of them are engaged as farm and family laborers and around 75% of female agricultural laborers do not get payment for their work. For gender mainstreaming in agriculture and the national economy, agricultural policy must ensure their right to own land and guarantee access to other productive resources such as credit, inputs, links to markets and agricultural extension services. The current study has not calculated female family members' services in household chores including food preparation for family and hired agricultural labor and therefore underestimates the female labor contribution. Future research may investigate female household services indirectly supporting agricultural laborers and the issue of agricultural labor substitution with machinery.

Wage-earning women are more efficient in resource use for production and spend more income on families compared to male wage earners and therefore indirectly contribute to achieving Sustainable Development Goal (SDG)#1, i.e., No Poverty, SDG#2, i.e., Zero Hunger, SDG#12, i.e., Responsible Consumption and Production, SDG13, i.e., Climate Action, as well as SDG#5 and SDG#8, i.e., Gender Equality and Decent Work and Economic Growth, respectively. The current study has attempted to capture the inter-household dynamics of female labor participation, but does not address intra-household agricultural labor dynamics and their relation to the decision-making process of family labor engagement. Understanding the intra-household labor participation dynamics will help us to understand the role of skills in constraining women in their labor force participation and the role of other contextual factors like culture and public support in the form of biased policies to discourage women's participation.

**Author Contributions:** I.M. provided the data for tables, conducted the interviews and all statistical analyses. S.A.A. supervised the research, M.A.K. helped with the research design, field survey and write up, S.-M.J. helped with the conceptual framework and overall study design, M.-u.-D.A. provided guidance about the dynamics of agricultural labor and the changing agricultural landscape in the cropping system, R.U. helped with the statistical results of the study, T.K. helped in selection of study area, identification of labor intensive crop stages, and in the revision of the draft. All authors reviewed the manuscript and revised it several times. All authors have read and agreed to the published version of the manuscript

**Funding:** The research was funded by the Australian government through its Sustainable Development Investment Portfolio (SDIP) and contributes to Australia's aid program.

**Acknowledgments:** This work was undertaken in collaboration with Commonwealth Scientific and Industrial Research Organization (CSIRO), which contributes to the Australian government's South Asia Sustainable Development Investment Portfolio and is supported by the Australian Aid program. The authors also sincerely acknowledge the thorough editing of the manuscript by Susan Cuddy of CSIRO, which greatly improved the quality of the paper.

**Conflicts of Interest:** The authors declare no conflict of interest.

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
