# Peer review of "Scale and Drivers of Female Agricultural Labor: Evidence from Pakistan"

_sustainability, doi:10.3390/su12166633_

Round 1

Reviewer 1 Report

Your article is very interesting and needed to know more about the situation of women working in agriculture in your country and offer guidelines to improve the situation. It is clearly written.

However, I think that some aspects can be improved. Here you have my recommendations:

Pay more attention to the sustainability consequences of this article. How the situation of women working in agriculture impact sustainability issues?

Stress more the key role of women that keep in the rural areas. If women keep in the women areas because they have incentives and can work or develop entrepreneurial activities, then the rest of the family will keep them too, as women are recognized the ones that apart from working in family agriculture, and starting entrepreneurial initiatives, they are the ones that take care of the family, ancient people, husband and kids. Add citations on this..

More information about the sample used in the article is required. What are the sociodemographic characteristics of men and women interviewed? What profiles did mainly answer the survey? Are women properly represented?

In the previous contacts with men and women to refine the questionnaire, what kind of questions were they asked? What questions were modified? Include this in your article, because it can impact in the theoretical framework.

I miss discussion and more practical implications.

After results, authors provide directly conclusions, however a discussion should be added between results and conclusions. To what extent your results agree with previous analysis on this or similar topics? What is new in your results? How can these results help to make decisions oriented to policy makers to improve the situation of women working in agriculture in these areas? You provide some insights or these practical implications in the conclusions but I think that they can be extended. Organizational and training incentives are also important.

In the conclusions, I also miss some limitations of the analysis and recommendations for future research from the theoretical and practical perspectives.

Author Response

Response to Reviewer 1 Comments

Point 1: Pay more attention to the sustainability consequences of this article. How the situation of women working in agriculture impact sustainability issues?

Response 1: Wage earning women are more efficient in resource use for production and spend more income on families compared to men wage earners and therefore indirectly contribute in achieving SDG#1 i.e. No Poverty, SDG#2 i.e. Zero Hunger, SDG#12 i.e. Responsible Consumption and Production, SDG13 i.e. Climate Action, besides SDG#5 and SDG#8 i.e. Gender equality and Decent Work and Economic Growth respectively.

Point 2: Stress more the key role of women that keep in the rural areas. If women keep in the women areas because they have incentives and can work or develop entrepreneurial activities, then the rest of the family will keep them too, as women are recognized the ones that apart from working in family agriculture, and starting entrepreneurial initiatives, they are the ones that take care of the family, ancient people, husband and kids. Add citations on this.

Response 2: The economic literature on female choice to stay out of job for domestic chores explains it through logic of lower opportunity cost for market activities and intra household welfare consideration and work dynamics [16]. Female access to productive and decent work is possible through removing religious and cultural constraints, efficient public provision of necessary services to reduce women time spent on household activities. and access to skills enhancement opportunities to enter the job market [17].

[16] Stratton, L. The Role of Preferences and Opportunity Costs in Determining the Time Allocated to Housework. The American Economic Review, 2012, 102(3):606-611.

[17] Singh, P., Pattanaik, F. Unfolding unpaid domestic work in India: women’s constraints, choices, and career. Palgrave Commun 6, 111 (2020).

Point 3: More information about the sample used in the article is required. What are the socio-demographic characteristics of men and women interviewed? What profiles did mainly answer the survey? Are women properly represented?

Response 3: The data were collected from 75 female and 225 male respondents. The number of female respondents is less because the majority of the randomly selected households either did not have family female participating in farming or the relevant female was not aware about managerial decision and female labor hiring related issues. However, to ensure objective response, the female enumerators triangulated the female labor related responses and double verified it from both male and female household members. The biasness of response in gender sensitive studies was known to the researchers and consciously both male and female family members were engaged in labor related questions to handle the issue of biased response and female representativeness appropriately.

Point 4: In the previous contacts with men and women to refine the questionnaire, what kind of questions were they asked? What questions were modified? Include this in your article, because it can impact in the theoretical framework.

Response 4: The initial expression of the researchers about female labor in agriculture was that the wage data would be easily available for different tasks. But due to the informal nature of the agricultural labor market and multiplicity of tasks performed, the questionnaire was later modified to indirectly measure wage and participation based on proportionate participation in different tasks.

Point 5:I miss discussion and more practical implications. After results, authors provide directly conclusions, however a discussion should be added between results and conclusions. To what extent your results agree with previous analysis on this or similar topics? What is new in your results? How can these results help to make decisions oriented to policy makers to improve the situation of women working in agriculture in these areas? You provide some insights or these practical implications in the conclusions but I think that they can be extended. Organizational and training incentives are also important.

Response 5: The finding that female labor contribution is considerably less in field activities is consistent with [20] observation that in South Asia fewer female work in the field. The finding about prevalence of female labor in livestock activities is consistent with findings of [10]. The study findings about employing family female or hired female labor for larger landholding size can be linked to findings of [21] that women labor is employed in agriculture when male labor is exhausted and larger farm size are first to exhaust male labor due to wide family male labor to land ratio. The role of women in the agricultural activities starting from seedling to post-harvesting of crops and managements of livestock has been reported by [19]. However, that study did not provide exact measures of time dedicated to each activity as provided by our current study. The role of education in female labor force participation in agriculture sector has not been previously investigated in detail. The existing literature on education and female labor force participation in agriculture is non-existent. The study results reflect that the relationship holds positive sign as far as hired female labor participation and education are concerned. The results are consistent with studies exploring relationship between education and labor force participation by [22-24]. A possible explanation for the positive role of education is that it boosts female confidence to work on farm as hired labor. However, the negative sign for family female labor with education could be due to intra-household labor dynamics resulting in non-participation of more educated family females compared to less educated family females.

Mellor, J.W. Agricultural Development and Economic Transformation: Promoting Growth with Poverty Reduction. 2017. Palgrave Studies in Agricultural Economics and Food Policy, Palgrave Macmillan.

Arshad, S., S. Muhammad, M. A. Randhawa, I. Ashraf, and K. M. Chuadhry. (2010). Rural women’s involvement in decision-making regarding livestock management. Pak. J. Agric. Sci. 47(2), 1-4.

Rehman, J. (2010). International human rights law. Pearson education, 56(2), 274-281

Pandey, R. and Pushpa Kumari, 2018. A review on gender discrimination faced by women in agriculture. IJRSR, 9(4C): 25674-25676

Sahn, D. E., & Alderman, H. (1988). The effects of human capital on wages, and the determinants of labor supply in a developing country. Journal of Development Economics, 29(2), 157-183.

Hafeez, A., & Ahmad, E. (2002). Factors determining the labor force participation decision of educated married women in a district of Punjab. Pakistan Economic and Social Review, 40(1), 75-88.

Faridi, M. Z., & Basit, A. B. (2011). Factors Determining Rural Labor Supply: A Micro Analysis. Pakistan Economic and Social Review, 49(1), 91-108.

Point 6: In the conclusions, I also miss some limitations of the analysis and recommendations for future research from the theoretical and practical perspectives.

Response 6: The current study has attempted to capture inter-household dynamics of female labor participation but did not address intra-household agricultural labor dynamics and its relations with decision making process about family labor engagement. The intra-household labor participation dynamics will help understand the role of skills in constraining women in labor force participation and the role of other contextual factors like culture and public support in form of biased policies to discourage women participation.

Reviewer 2 Report

The paper respect the requirements of an academic research article.

The state of Pakistani female working in agriculture is analysed on different types of crops, farm size, and type of employment (hired or family). The area of the study is in Pakistan, Punjab province. The data consisted in 300 households from the above mentioned province.

The literature review could be improved by presenting similar study results from around the world, or techniques used to study the problem.

The data acquisition and research framework is well presented and suited for the studied problem.

The conclusions are supported by the results.

The study presents a detailed picture of female status working in agriculture and the gender gap specific to this part of the world. The independent variables included in the study are significant and pertinent for the studied problem. The study has practical importance  as on its basis policy makers could legislate new reforms to reduce the gender gap.

Author Response

Response to Reviewer 2 Comments

Point 1: The literature review could be improved by presenting similar study results from around the world, or techniques used to study the problem.

Response 1: For example, [13] analysed a sample of 363 households collected from 4 villages in Iran and found positive attitude of female farmers towards agriculture sustainability. Another study collected data about agricultural labor from 500 farmers in South Punjab, Pakistan and found that the majority of livestock activities were performed by female laborers [14]. The social and cultural barriers hinder effective female participation in agriculture and other economic activities were investigated through resistance to education and other learning activities [15]. The economic literature on female choice to stay out of job for domestic chores explains it through logic of lower opportunity cost for market activities and intra household welfare consideration and work dynamics [16]. Female access to productive and decent work is possible through removing religious and cultural constraints, efficient public provision of necessary services to reduce women time spent on household activities. and access to skills enhancement opportunities to enter the job market [17]. While there are several other studies on gender labor participation and socioeconomic barriers, there is a dearth of literature on gender labor relations in family and hired labor and its determinants.

Ezatollah, K. (2007). Sustainable agriculture: towards a conflict management based agricultural extension. J. Appl. Sci. 7(24), 3880-3890.

Hassan, M. G., and M. Leach. (2010). Demand response experience in Europe: Policies, programmes and implementation. Energy, 35(4), 1575-1583.

Luqman, M., R. Saqib, X. Shiwei, and Y. Wen. 2018. Barriers to gender equality in agricultural extension in Pakistan: Evidence from District Sargodha. Sarhad J. Agric.  34(1):136-143

Reviewer 3 Report

The authors of the article take up the rarely discussed problem of women's work in agriculture in Pakistan. I have one comment concerning the theoretical introduction: if possible, it would be useful to present the situation in Pakistan in the wider context of other countries, eg India and/or Afghanistan (this note relates to the presentation of research and/or statistics on this subject).

Summarizing, the research has been conducted in an appropriate manner and the paper is prepared at a good level. Presented study is an interesting introduction to future research.

Author Response

Response to Reviewer 3 Comments

Point 1: I have one comment concerning the theoretical introduction: if possible, it would be useful to present the situation in Pakistan in the wider context of other countries, eg India and/or Afghanistan (this note relates to the presentation of research and/or statistics on this subject).

Response 1: 

Ezatollah, K. (2007). Sustainable agriculture: towards a conflict management based agricultural extension. J. Appl. Sci. 7(24), 3880-3890.

Hassan, M. G., and M. Leach. (2010). Demand response experience in Europe: Policies, programmes and implementation. Energy, 35(4), 1575-1583.

Luqman, M., R. Saqib, X. Shiwei, and Y. Wen. 2018. Barriers to gender equality in agricultural extension in Pakistan: Evidence from District Sargodha. Sarhad J. Agric.  34(1):136-143.

Sahn, D. E., & Alderman, H. (1988). The effects of human capital on wages, and the determinants of labor supply in a developing country. Journal of Development Economics, 29(2), 157-183.

Hafeez, A., & Ahmad, E. (2002). Factors determining the labor force participation decision of educated married women in a district of Punjab. Pakistan Economic and Social Review, 40(1), 75-88.

Faridi, M. Z., & Basit, A. B. (2011). Factors Determining Rural Labor Supply: A Micro Analysis. Pakistan Economic and Social Review, 49(1), 91-108.

 Arshad, S., S. Muhammad, M. A. Randhawa, I. Ashraf, and K. M. Chuadhry. (2010). Rural women’s involvement in decision-making regarding livestock management. Pak. J. Agric. Sci. 47(2), 1-4.

Round 2

Reviewer 1 Report

You have properly worked on my recommendations and suggestions.